# An Overview of Urban Mobility: Revolutionizing with Innovative Smart Parking Systems

Gaetano Rocco *,†, Claudia Pipino † and Claudio Pagano †

Department of Business Science-Management and Innovation System, University of Salerno, 84084 Fisciano, Italy; cpipino@unisa.it (C.P.); cpagano@unisa.it (C.P.)
* Correspondence: garocco@unisa.it
† These authors contributed equally to this work.

**Abstract:** This feasibility study aims to propose the design of a smart parking architecture that is able to offer new services by exploiting the latest IoT technologies. This innovative solution is designed for use by both public authorities and private individuals who need to manage urban parking areas efficiently. The contribution of this work is to attempt to define the requirements and technical choices that can be made for the design of a system that adheres to the paradigm of innovation and efficiency in smart parking. Indeed, there is a practical limit between the existing models and the best solutions to apply. For each technology, the following details are provided: the problem to be managed, the current state of the art on the market, the main solutions, and also the related commercial policies. We proceed with the "Outline Design", which integrates the technical specifications and defines the main information flows between the functional blocks. The results of experimentation show that the proposed reservation-based parking policy has the potential to simplify the operations of parking systems, as well as alleviate cities' traffic congestion.

**Keywords:** smart city; smart parking; IoT; ICT

## 1. Introduction

The concept and role of smart cities as a new innovative paradigm for urban planning and management have been the subject of particular attention in recent years [1,2]. The smart city can be defined as a place where traditional networks and services provided to citizens and firms are improved through technological innovation by fostering environmental and social well-being [3,4]. In particular, a smart city can be seen as an ultra-modern urban area that responds to the needs of businesses, institutions, and, above all, citizens [5]. In fact, it is possible to generate new value and offer better services for citizens within smart cities thanks to the advances offered by new ICT technologies [6,7]. In smart cities, there are various contexts and dynamic changes that create complexity and uncertainty in the design of city services. The lack of contextual analysis can contribute to this when understanding how a city should be prepared to accommodate various contexts. Therefore, this intelligent environment can be considered following the vital systems approach [8,9], analysing city services as interconnected systems that influence each other. So, it is very important to understand how they interconnect in order to instantiate communication with each other [10]. Also, in the design of complex services, it is important to consider that stakeholders operate within various contexts, as many cases involve multiple value proposition service chains. The implementation and design of these intricate services cannot be isolated from other services within the entire ecosystem. One of the most important roles in service modelling is the agent, who can be the provider, the contributor, or the recipient of the service. How individual agents and teams are organised in the value proposition chain is a key factor in increasing the efficiency of the service [11]. With the rapid growth of Industry 4.0-enabling technologies, various applications in different domains of smart

cities, such as smart city planning or smart mobility, can be realised for the design of smart services. However, understanding the interconnections and mutual influences between city services across application domains is still very complex [12]. So, it is very important to understand which services are most used and which is the best offering to efficiently support citizens in a data-driven world, and therefore a lot of data are needed. So, it results very important to understand which services are most used and which is the best offering to efficiently support citizens in a data-driven world, a lot of data is needed [13]. A smart city requires an adequate ITC (Information and Communication Technology) infrastructure to operate efficiently. This infrastructure is made possible by technologies like big data, Internet of Things (IoT), and cloud computing, which utilise advanced sensors, measurement and analysis tools, and solutions based on machine learning and artificial intelligence to generate value for citizens [14]. IoT plays a significant role in various aspects of a smart city citizen's life, including health, security, and transportation. It also has implications at the national level concerning policy decisions related to energy conservation, pollution reduction, remote monitoring, and necessary infrastructure. By leveraging IoT, smart cities can achieve more efficient, economical, and secure operations based on different factors such as energy-saving policies, economic considerations, and reliability levels. However, one prominent challenge faced by people in smart cities is the scarcity of car parking facilities and traffic management. In busy cities, finding a free parking slot during peak hours can be extremely difficult, leading to increased traffic congestion [15]. Traffic congestion in and around large cities has led to significant time wastage and the exacerbation of traffic-related issues. This includes fuel wastage, driver frustration, and increased air pollution. Consequently, this leads to the elevation of emissions such as Carbon Monoxide (CO), Carbon Dioxide (CO2), Volatile Organic Compounds (VOCs), Hydrocarbons (HCs), and Nitrogen Oxides (NOx), thereby contributing to air pollution. The United Nations Environment Program (UNEP) reported that approximately 7 million premature deaths globally are attributed to air pollution [16]. To address these issues, cities can create a common ecosystem that connects service providers, users, infrastructures, and communities. This cooperation is essential for effective governance of the territory and the overall well-being of citizens. Two cities, Treviso and Mantua, in Italy, have served as examples of successful smart parking initiatives. Treviso has been a pioneer in smart parking since 2009 with its I-Park system, which resulted in increased revenue for the municipality [17]. Mantua, in partnership with private companies, the Public Administration, and the University of Modena and Reggio Emilia, launched a pilot project in 2019 that transformed city stalls into intelligent parking lots. Through the use of IoT and 66 parking sensors, Mantua's project led to a decrease in traffic, pollution, and time wasted searching for parking spaces (https://www.unimore.it/, accessed on 20 July 2023). The Bosch project in Mantua is a notable initiative. It involves the collection of various data, which the municipality processes in collaboration with the University of Modena and Reggio Emilia. Thanks to the wireless infrastructure, information is seamlessly transmitted, allowing the sensors installed in the road to communicate with the central database. This kind of technology, similar to what has already been successfully implemented in Treviso, contributes to identifying parking abuses and efficiently monitoring spaces reserved for electric vehicles. The benefits of these smart city projects extend beyond just parking solutions. IoT technologies, when integrated into various aspects of urban life, have the potential to improve citizens' well-being and overall quality of life. For example, IoT can be utilised to enhance public health services, optimise public transportation systems, and enhance the security and safety of the city. By leveraging big data and advanced analytics, city authorities can make data-driven decisions to address challenges like energy consumption, pollution, and traffic congestion. Moreover, the ability to monitor and manage different urban services remotely can lead to more efficient resource allocation and cost savings for the municipality. As the development and implementation of smart city technologies continue to evolve, it is crucial for cities to learn from successful experiences like those in Treviso and Mantua. Collaboration between the public sector, private companies, and academic institutions is vital to drive innovation and

achieve sustainable smart city solutions [18]. The smart city technologies, enabled by IoT and other advanced tools, offer numerous benefits ranging from improved urban services and resource management to enhancing citizens' everyday lives. The successful cases of Treviso and Mantua demonstrate the potential of these technologies in making cities more efficient, sustainable, and enjoyable places to live.

The research question that this study will try to answer is the following:

What are the key technical requirements and design considerations for implementing a reservation-based smart parking architecture to optimise urban parking management and alleviate traffic congestion?

## 2. Generic Requirements of Smart Parking System

Within the context of smart cities, one of the most significant applications of IoT technology is the smart parking system (SPS) [19]. The central concept behind this system is the automated allocation of parking slots without the need for human intervention [20]. SPSs employ IoT technology and devices, such as sensors, to assist drivers in locating the nearest available parking spaces, thereby reducing traffic congestion, air pollution, and associated health risks. The "smart parking" system is designed to provide real-time information about parking space availability in a specific area, accessible through a mobile application (app) [21]. Users can view available parking slots on an interactive map, initially from a macro perspective. Once a specific area of interest is selected, users can access a more detailed "planimetric" view that displays a list of contiguous and non-contiguous parking spaces, making it easier for them to choose an available spot [22]. The implementation of smart parking systems not only benefits drivers by saving time and reducing stress in finding parking, but also contributes to creating a more sustainable and eco-friendly urban environment by optimizing the utilization of available parking spaces and minimizing unnecessary traffic movements. By incorporating IoT technology into smart city initiatives such as smart parking systems, cities can enhance the overall efficiency of urban transportation, decrease the environmental impact of vehicles, and improve the overall quality of life for their citizens. With technology's continued advancements, we can anticipate even more innovative solutions to address urban challenges and shape the cities of the future. Finding parking spaces in metropolitan areas, especially during rush hours, poses a challenging task for motorists primarily because they lack real-time information about available spaces. Nowadays, smart parking systems (SPSs) help alleviate this issue by assisting drivers in locating free parking spaces, offering personal benefits such as time and cost savings, as well as societal benefits such as reduced pollution and city traffic congestion. However, these systems are not without their flaws [23]. One issue is the possibility of multiple drivers heading for the same free parking spaces simultaneously, leading to unintended competition. Consequently, drivers unable to secure a parking space might end up continually changing their driving habits in search of new parking spots, creating a frustrating cycle [24]. Moreover, some applications may transmit the same information about free parking spaces to several users in the vicinity, potentially causing imbalances in parking space usage. This imbalance could result in certain parking spots becoming overcrowded while others remain underutilised, leading to traffic congestion problems for city traffic authorities. Additionally, the constant need for users to monitor updates on their smartphones while searching for parking spaces through the app can pose safety risks, diverting their attention from the road and potentially causing accidents involving their vehicles, other cars, or pedestrians [25]. To enhance the effectiveness of smart parking systems, it is essential to address these challenges. City traffic authorities and developers of such systems need to implement solutions that distribute information in a more balanced manner, preventing congestion in certain parking areas. Moreover, focusing on user-friendly interfaces and minimizing distractions on smartphones can help ensure the safety of drivers and pedestrians. By continually improving and refining smart parking systems, cities can fully leverage the benefits of IoT-based technology to enhance urban

mobility, decrease environmental impacts, and improve the overall driving experience for their citizens.

## 3. Context Analysis

It is necessary to analyse the technologies available or in the process of consolidation on the market that appear to be the most promising for the creation of a "smart parking" system [26]. The technologies analysed in terms of formalising the technical problem to be managed, the reference context, and the solutions available on the market are as follows:

- IoT (Internet of Things) communication networks;
- Localisation systems;
- Ultra-low-power sensors;
- Online booking systems.

IoT communication networks: One of the main assets of the smart parking system is the installation on the stalls of parking sensors that detect whether the stall is free or occupied. They are able to communicate any status changes in real time to the centralised platform so users looking for parking will be addressed soon and correctly [27]. One of the main problems to be faced in this context is equipping the sensor with a very-low-consumption wireless battery as the only energy source. In order to connect devices in the field using very-low-consumption wireless communication modules, IoT networks have been proposed on the market, whose technological, application, and performance characteristics are analysed in the following paragraphs to define their usability in the smart parking system. The IoT represents an aggregation of disparate devices that, independently, produce and/or receive data using the extreme layers of the network protocols. Therefore, it is important to carefully evaluate the limits and technologies applicable to create communications systems and networks that involve IoT systems. The Internet of Things combines personal area networks, local area networks (LAN), and long-range wide-area networks (WAN) in a unique network of communication channels, making it possible to exchange information [28]. The complexity and articulation of an IoT network require the definition of a complex communication network that allows for better adaptation to the particularities of the project. On the basis of the distances to be covered, the use cases, and the transmission speeds to be guaranteed, numerous communication protocols and architectures can be hypothesised. The acronym WPAN (Wide Personal Area Network) is often used in conjunction with other acronyms related to near-range communication systems, such as wireless Field Area Network (FAN), Wireless Local Area Network (WLAN), wireless Home Area Network (HAN), wireless Neighbourhood Area Network (NAN), and Wireless Body Area Network (WBAN). Most IoT networks operate in a frequency range centred on 900 MHz and 2.4 GHz. The main characteristics of these two frequency bands are summarised in Table 1 below.

**Table 1.** Characteristics of 900 MHZ and 2.4 GHz frequencies.

| Characteristic | 900 MHz | 2.4 GHz |
| --- | --- | --- |
| Signal strength | Generally reliable | Crowded and interference-prone area of the electromagnetic spectrum |
| Distance covered | Approximately 2.6 times greater than that covered by 2.4 GHz networks | Shorter, however compensable with improved encoding |
| Penetration | Long node can penetrate most materials and vegetation | Potential interference with building materials |
| Data rate | Limited | About 2/3 times greater than 900 MHz |
| Signal interference | High obstacles can attenuate the signal, it propagates better in foliage | Less chance of interference on some channels |
| Channel interference | Possible interference with cordless phones, RFID scanners, etc. | Interference with 802.11 WiFi standards |
| Costs | Medium | Bass |

From the analysis shown in the table, the choice of the band depends on the operating context for which in the rest of this section we analyse some of the market solutions, highlighting the technological choices made, and their positive and negative aspects. Localisation systems: One of the main services that the smart parking app intends to provide is the real time guidance of the user in a precise manner towards the parking spaces available in the area of interest [29]. To provide this service in a qualitative way, the support of a localisation system is required, capable of ensuring a high degree of precision even in highly disturbed contexts such as urban ones (road canyons). The most widespread localisation technologies and their applicability to the reference context will be analysed. Geolocation is the process by which it is possible to relate certain information to a specific point on the Earth's surface, identified using localisation methods (therefore the retrieval of the co-ordinates of the latitude and longitude) of an electronic apparatus, exploiting its physical characteristics or receiving detailed information from the tool itself. Specifically, the concept of geolocation emphasises the dynamic aspect, referring to information that can frequently be updated, such as position and speed. The possibility of retrieving such information dynamically and detecting it in more detail is derived from the combination of different technologies that have been developed in the twenty-first century and now give each device the possibility to communicate it in real time. Thanks to the capacity to immediately correlate the information to a single device and its geographical position, it is considered one of the most revolutionary development fields in the social and economic sphere [30,31]. All localisation systems consist of two phases: data retrieval and processing. The main systems that characterise information retrieval in external (outdoor) and internal (indoor) environments are presented below. In this section, we will discuss the methods used to calculate the position of mobile devices based on the type of data that we managed to obtain. The methods used to estimate the position are as follows: (1) Arrival angle (AOA) is where the position is determined by the direction of incoming signals sent by other transmitters whose position and directional signal is known. A triangulation technique is then used to calculate the position of the device. (2) Cell identity (CI) is where the identification of the cells is read and the position close to that of the transmitting cell is considered. This method is very imprecise, especially if connected to a cell with a very wide range of action. In addition to use via a mobile phone, this technique can also be used indoors by calculating the position from the connection, or lack thereof, of the device to a signal source. (3) Time of arrival (TOA) measures the round trip in time of the signal, triangulating it on the available antennas. The accuracy is very high, but it strongly depends on the accuracy of the devices' clocks. (4) Signal strength uses attenuation properties due to signal propagation to determine the mobile device's location. These methods are the most common in outdoor environments but can also be suitable for indoor situations. On the other hand, the location fingerprinting technique is almost exclusive as far as restricted environments are considered. This technique is relatively simpler than the arrival angle (AOA) or arrival time (TOA) ones, and can be performed without specialised hardware, simply by reusing an existing LAN network. Table 2 summarises the main systems for detecting, locating, positioning, and tracking objects, people, and vehicles in real time (RTL (Real-Time Location)) that are currently on the market or under development.

**Table 2.** Summary of main detection systems.

| (a) | | | |
|---|---|---|---|
| Technology | Tag | Operation | Coverage |
| GPS | Ric. GPS | Constellation of 24 transmitting satellites from which the tag calculates the $\langle x, y, z \rangle$ triples | Outside |
| D-GPS | Ric. DGPS | GPS with decreased error by means of one or more stations placed on the ground | Outside |

**Table 2.** *Cont.*

| (a) | | | |
|---|---|---|---|
| **Technology** | **Tag** | **Operation** | **Coverage** |
| Cell-ID | Sim Tag or mobile phones | The cell to which the phone or tag is connected is identified | Indoor/outdoor |
| Telecom | Sim Tag or mobile phones | Cell-to-Cell Triangulation (BTS) of telephone operators | Indoor/outdoor |
| Wi-Fi | Pc Wi-Fi or Tag Wi-Fi | Triangulation between 3 or more access points (Tdo = Time Difference of Arrival) | Wi-Fi AP coverage zones (minimum 4) |
| RFID Passive | Passive tags HF/UHF | A tag transmits its ID when it enters the output range of a reader | In the presence of RFID reader |
| RFID Active | Tag Active RF | The tag continuously emits its ID and is located by the receiver in its field | Reception area of receivers |
| UWB | UWB Active Tags | The tag emits continuously: at least 3 receivers measure the time difference | Coverage area of $\frac{3}{4}$ receivers |
| ZigBee | Active Tags ZigBee | Mutual triangulation between tags (mesh network) with at least 4 tags in known position | Areas where at least 4 tags in a fixed position can see moving tags |
| (b) | | | |
| **Technology** | **Accuracy** | **Benefits** | **Disadvantages** |
| GPS | 1 m–20 m | High dissemination | Battery consumption, high startup time, subject to Mil. USA |
| D-GPS | 1 m–5 m | High accuracy | High costs, needs ground stations |
| Cell-ID | 50 m–1 km | Diffusion of mobile phones | Low accuracy |
| Telecom | 20 m–200 m in the city | Diffusion of mobile phones | Data not released by Telecom |
| Wi-Fi | 2 m–5 m | Integration position + data transmission | Sensitive to noise and reflections |
| RFID Passive | 1 cm–10 m from the receiver | Low tag costs | Localization only in proximity of receiver |
| RFID Active | 1 cm–100 m from the receiver | Robustness of the signal | Low accuracy, high-cost tags |
| UWB | 10 cm–30 cm | Resistance to reflection (multipath) | Non-global standards |
| ZigBee | 50 cm–1 m | Simplicity, extendibility | Poor tag availability |

Ultra-Low-Power Sensors: A core feature of the smart parking platform is the ability to detect the status of a stall (free/occupied) as accurately as possible and to transmit it in real time to the platform to use this information in presenting the user with the parking offer in a certain area. For solutions that plan to install distributed systems in the field such as sensors, it is also required that they are of the ultra-low-power type to guarantee energy autonomy for many years and avoid wiring that would severely limit their use. Therefore, the main technologies used for detecting the stall state are analysed as well as a more in-depth examination of the most suitable ones for the project's purpose [32]. The detection of the presence of vehicles in the stalls is essentially based on two types of sensors: (1) fixed, which operate by detecting the presence or absence of vehicles by updating information in a short period of time and (2) furniture, which can detect the state of occupancy when the car passes through special portals at the entrance of the car park, updating the information with longer times. Different sensors have distinct ways of detecting the presence of vehicles. Infrared sensors: The passive infra-red sensor receives radiated heat from the human body or vehicles and is often used in conjunction with other sensors to tell if a driver gets out of the car after parking. The active infrared sensor measures the distance to any obstacles ahead. The infrared sensor is very sensitive to the sun and any kind of environmental object so the detection accuracy is not so good.

Ultrasonic sensors: Similar to the infrared sensor, the ultrasonic sensor uses sound instead of light to make measurements, presenting better performance in outdoor environments. The ultrasonic sensor provides a more complex signal pattern with multiple detection possibilities when used in both stationary and mobile scenarios [33]. Acceleration sensors: Acceleration measurements mainly work by measuring the instantaneous vibrations of the ground produced by moving vehicles [34]. Since vehicles are usually the heaviest objects travelling on the road, the accelerator can infer that a vehicle has arrived at its destination and park with the help of other sensors, such as the optical sensor. An optical sensor can, for example, be installed where the light can be dimmed from a parked vehicle. However, the optical sensor is vulnerable to any light source and transient, permanent objects; therefore, its accuracy is still questionable. Magnetometers: The magnetometer is the most popular stationary parking sensor, particularly for municipal distribution. It measures the magnetic field and detects the arrival of large metal objects; its signal pattern is easy to read and accurate, but does not support multiple detection. It is also more expensive than the sensors mentioned earlier. Alternatively, it is possible to install the magnetometers along the obligatory path and compare the different counts between two adjacent sensors in order to know approximately how many vehicles are parked between the stalls. Other types of sensors: Cameras provide a much more complicated signal model than ultrasound. Both require sonar images to be heavily processed in order to extract the desired information from the background noise. Despite the complexity of the processing, they have sparked research interest due to some additional information, related to crime scenes or personal privacy, that these sensors could retrieve. Other sensors of interest are inductive and piezoelectric ones. Both operate through contact and are installed on the road surface. The inductive loop is a mature and widely used technology for traffic surveillance, and it simply detects whether a vehicle is passing through the measurement area on an inductive loop embedded in the road surface. The piezoelectric sensor is similar to the inductive loop but is able to read more information from the pressure exerted on it. This type of contact sensor requires intrusive installation and is easily worn out in cases of frequent use.

Online Booking Systems: The basic idea for the "smart parking" project is that the user, when they need a parking space, in addition to being informed in real time where the free parking spaces are, can also use a booking service to "block" the parking space.

The allocation of a certain number of parking stalls explicitly to a reservation service at a higher cost compared to regular parking has become crucial in complex urban settings, such as large cities, where thousands of stalls need to be dedicated to reserved parking with a high daily turnover. In response to this need, an online booking system is essential, which should possess the following capabilities:

- Accurately model the availability of free parking spaces within a booking service.
- Efficiently manage a large volume of booking requests made through various applications.
- Offer seamless integration with other platform modules that contribute to the overall service provision, including browser and payment functionalities.
- Maintain a pricing structure commensurate with parking costs.

Conducting a market analysis is essential to assess whether existing online booking platforms meet the minimum criteria for integration into the smart parking platform. The online booking softwares serves to streamline booking and reservation processes for customers, staff, and agents, demanding high levels of reliability and efficiency. The main advantage of utilizing a booking platform is the real-time accuracy and convenience it offers to customers. Such platforms provide up-to-date booking information, secure payment mechanisms, and other automated features that expedite the booking procedure. To explore the online booking software landscape, researchers can evaluate popular platforms through free or demo trial plans commonly offered by software providers. Additionally, experts' evaluations [1,19], consisting of thorough reviews based on industry standards and user satisfaction ratings, can aid in the decision-making process. An objective criterion for assessing the suitability of online booking software is whether the business operates an

online portal (e.g., company website or social media page) and handles a significant number of bookings. In such cases, adopting an online booking software becomes indispensable for efficiently managing the entire process. The software should possess intuitive features for customers to access information, updates, and support; complete fillable forms; view reservation timetables; and process payments. Since booking operations significantly contribute to revenue generation, ensuring flawless and reliable online booking software is of utmost importance, preventing customers from seeking alternative booking services due to system glitches. When implementing an online booking system (OBS) for smart parking, several important factors should be taken into account:

1. Internet Access: As with all online computing systems, uninterrupted Internet access is essential. However, potential disruptions such as power failures, hardware or software failures, or locations with no internet connection must be considered. A reliable backup system should be in place to ensure continuous service availability.
2. Data Security: Data security is paramount, even for the most secure systems. Whether data are stored in the cloud or on a central corporate server, access should be restricted to authorised personnel only. Regularly updated authentication protocols must be implemented to safeguard sensitive information from attacks and breaches.
3. Technical Support: Timely technical support is crucial, especially during critical situations. System failures can occur, and periodic maintenance is necessary for optimal performance. Ensuring that the service provider offers immediate technical support is vital for seamless operations.
4. Integration: Smooth integration with existing office or business applications is essential for efficient workflow. The OBS application should seamlessly work alongside other applications, reducing the need to switch between software for different tasks.
5. Mobile Payment Systems: As part of the smart parking project, a comprehensive flowchart is planned, providing users with information on available parking spaces and enabling automatic and digitised payment for the service. The mobile payment platform must effectively manage the modelling of parking service offers, considering variations between cities and different areas within a city.

The service offer provides the possibility of activating extra incremental paid services. For instance, a basic service may indicate the area with free stalls, and additional services can be activated for a fee based on the In-Thing Purchase logic. This allows users to navigate to the free stall and book it in advance. The system must handle massive user profiling, potentially dealing with millions of users, manage micropayments, and provide response times appropriate to the mode of service usage. Mobile payment is a service enabling payments or money transfers via smartphones. It allows transactions to be debited from various payment instruments, including cash, to virtual purses, but not telephone credit unless facilitated by a payment institution (PI) or electronic money institution (EMI). The service utilises wireless communication networks such as GSM, UMTS, Bluetooth, RFID, and the mobile phone itself as a physical tool to activate the payment. Mobile payment encompasses multiple usage paradigms [35], including the following:

- Mobile Remote Payment [36]: payment made remotely through mobile devices.
- Mobile Commerce [37,38]: conducting commercial transactions via mobile devices.
- Mobile Money Transfer [39,40]: transferring money electronically using mobile devices.
- Mobile Proximity Payment [41]: payment made in close physical proximity using mobile devices.

### 3.1. Functional Block Architecture and Technologies

The functional block architecture allows you to conceptually have a general over-view of the operation of the various application software and how they interact with each other. Each software module, in fact, is connected to another by data flows that diversify its operation. Figure 1 displays the functional block architecture of the smart parking system, wherein each functional block represents a distinct software or hardware module.

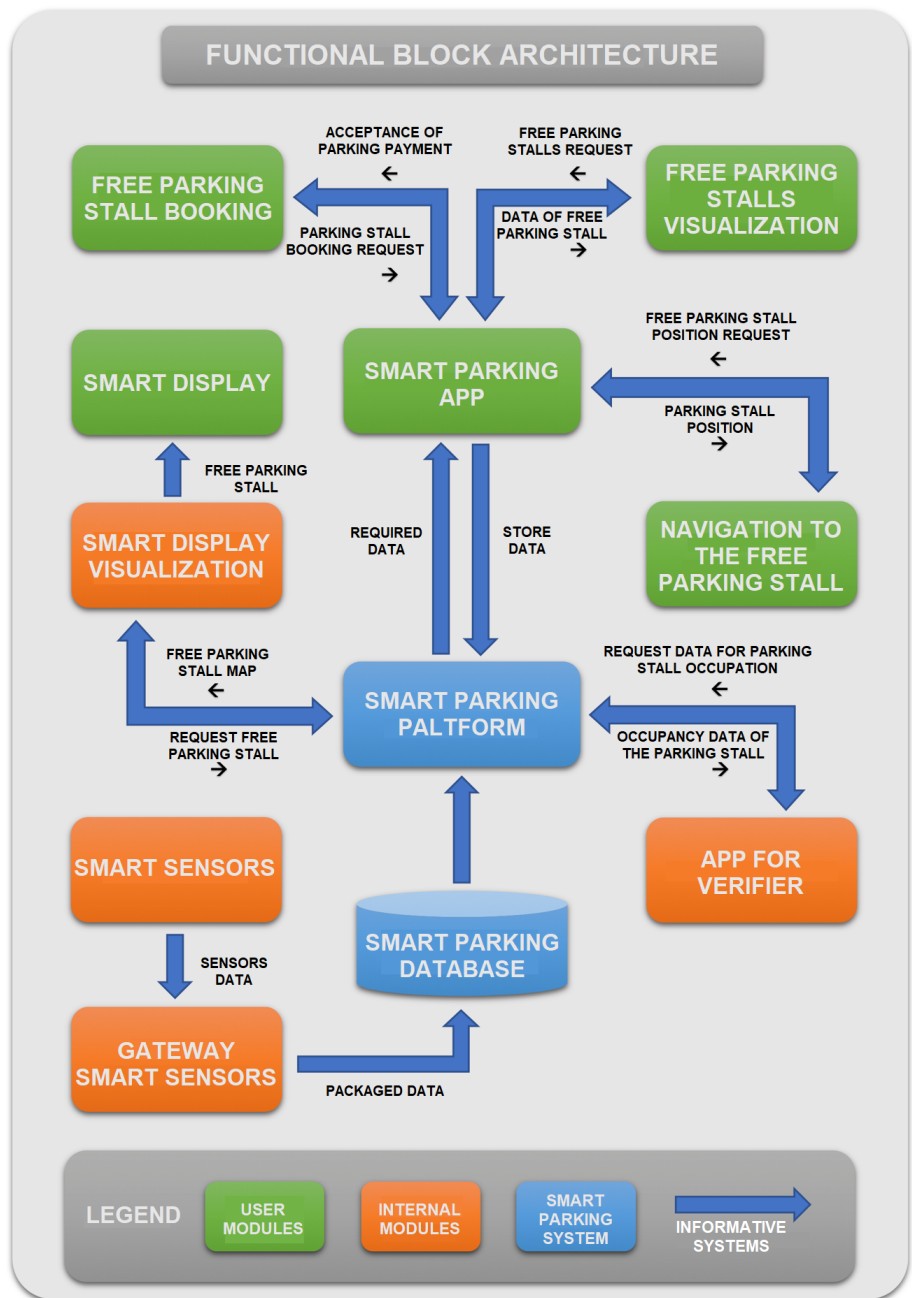

**Figure 1.** Functional block architecture of the smart parking system.

As shown in Figure 2, the architectural components illustrate the different elements that constitute the smart parking system, along with the proposed technologies recommended by the feasibility study for its implementation.

In particular, the *Smart Sensor* is a hardware module comprising a sensor positioned at the centre of each parking stall, embedded in the subsoil. These sensors fall under the category of magnetometric sensors, capable of accurately detecting variations in the magnetic field caused by the movement of a ferromagnetic vehicle on the stall [42]. When a vehicle parks on the sensor, the magnetic flux variation is recorded and transmitted to the reference gateway through a LoRaWAN infrastructure. Equipped with a low-to-medium-gain antenna, the sensor can receive and transmit radio waves in the 868 MHz spectrum, assigned for the European region. With this technology, the sensor can communicate with its respective reference gateway at up to 2.5 km in urban environments and up to 20–25 km in rural areas. Each parking stall is numbered and geolocated at the time of

sensor installation, optimizing their use, facilitating traceability on maps, and ensuring easy navigation through the app. The sensors will have a minimum autonomy of 10 years, achieved using remote rechargeable batteries with wireless charging systems.

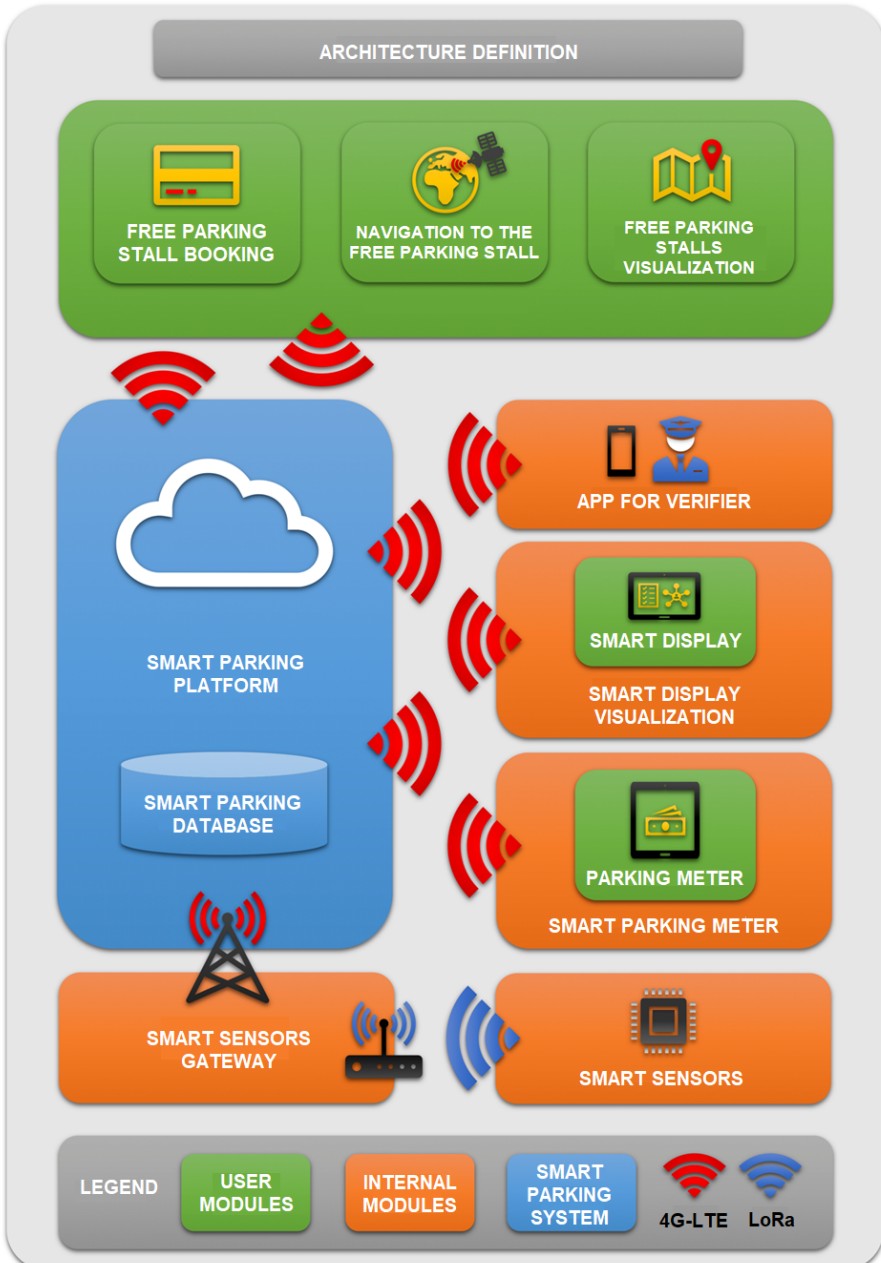

**Figure 2.** Proposed smart parking system.

The *Smart Sensor Gateway* is another hardware module responsible for receiving data from multiple sensors integrated into the asphalt of parking stalls. Its primary function is to collect the input data from sensors and package them for transmission to the *smart parking* reference platform. Essentially, the gateway acts as a router equipped with a radio wave concentrator that receives and transmits data from sensors in adjacent underground stalls. The routers are multi-channel devices, capable of real-time data acquisition from multiple sensors, managing up to 128 Long-Range (LoRa) nodes simultaneously. Once all the reference gateways collect the data, they are sorted and packaged into a single file and sent to the cloud (network server) through infrastructures based on 4G-LTE telecommunication technologies. The *smart parking* platform is the software module that receives the data

transmitted by the *smart parking* database, which serves as the hardware "layer" of the platform. The *smart parking* app module allows users to view available parking spaces, book a parking spot, make payments for the parking service, and report to public and/or private entities responsible for the service provision. Payment for the service can be made through various online payment channels such as PayPal (https://www.paypal.com, accessed on 20 July 2023), credit cards, and PagoPA (https://www.pagopa.gov.it, accessed on 20 July 2023), which also offers the option of recharging an electronic purse for paying the service. Through the portal, users can check the contents of their electronic purse, reload it, download invoices, and review their reservation and payment history. The *Verifier App* module enables inspectors to access essential information for validating parking occupancy. It allows inspectors to check the parking time and date, car license plate, and whether parking has been extended or has expired. In case of violations, the app enables a sanction to be issued with a warning print that can be placed on the car's windshield wiper. Communication between the *Verifier App* and the *smart parking* platform occurs exclusively through 4G-LTE telecommunication networks. For the development of the *Verifier App*, the Ionic Framework (https://ionicframework.com, accessed on 20 July 2023) will be utilised, which is a widely used open-source SDK for building hybrid mobile applications. It is built on Angular (https://angular.io, accessed on 20 July 2023) and Apache Cordova (https://cordova.apache.org, accessed on 20 July 2023) technologies, which are frameworks for developing web and mobile applications. The *smart parking* platform, constituting the back-end part responsible for facilitating interactions between various software modules, will be implemented using the Spring Framework https://spring.io, accessed on 20 July 2023), based on Java (https://www.java.com, accessed on 20 July 2023) technology. *Smart parking meter*: This software module allows users without an app to pay for the parking service by entering the vehicle's license plate number occupying the stall. Payment can be made through the main online payment methods using the parking meter user interface. After verifying that the stall is free, the parking meter communicates its effective occupation to the *smart parking* platform, which confirms the successful completion of the operation as soon as the receipt is issued. All checks are performed electronically through the *Verifier App*. For the development of this application, we will use the Ionic Framework, widely used for implementing web and mobile applications. Communication will exclusively take place with the *smart parking* platform module via LoRaWAN telecommunication networks.

Parking Meter: It is equipped with a solar panel and battery buffer that ensure continuous operation for at least 16 h a day in the absence of connection to the electricity grid. It uses LORA communication modules and ultra-low-consumption light-emitting diode (LED) displays to facilitate the continuity of operation.

Smart Display View: This visualization software module allows the map and information received from the free parking spaces to be sent to displays in areas adjacent to them. It communicates unidirectionally with the various displays distributed throughout the territory and with the *smart parking* platform in both directions. For the development of this application, we will use the Ionic Framework, widely used for implementing web and mobile applications. Communication will exclusively take place with the *smart parking* platform module via 4G-LTE telecommunication networks.

Smart Display: It is a system for displaying information about the number of free parking spaces in a nearby car park. These displays will always be connected to the *smart parking* platform through the Smart Display Visualization software module and will use a technology that significantly reduces consumption while preserving the clarity and ease of reading information by motorists or users in general. The displays will use E-Ink technology based on the physical process of electrophoresis. As explained in the previous market analysis, this technology allows better visibility to users even from far distances and drastically reduces consumption, as only the change in status requires a small energy consumption. The display is constantly connected to the reference platform, which sends the updated status of the stalls to the display. The information to be displayed is received by means of small hardware modules that operate with technologies based on NB-IoT

telecommunication networks. The display will be equipped with a solar panel of adequate size and buffer batteries that will ensure a minimum autonomy of 16 h a day for at least 7 years to ensure energy autonomy.

*3.2. Technical–Economic Feasibility*

The general plan of the activities necessary for the creation of the *smart parking* platform is implemented in the subsequent Phase 2. The plan is organised into activities and tasks briefly presented in the list and subsequently described in greater detail.

Activities ($A_x$) and Tasks ($A_{xy}$) list:

A1—Detailed design

- A1.1—Detailed design of the system
- A1.2—Creation of the development and test environment in the laboratory

  A2—Realization of the field infrastructure

- A2.1—Stall occupation detection sensors;
- A2.2—Information panels on the street;
- A2.3—Parking meters.

  A3—Development of the "smart parking" cloud application

- A3.1—Development of software modules;
- A3.2—Integration and testing in the laboratory.

  A4—Pilot and recommendations

- A4.1—Agreements with the pilot and pilot design;
- A4.2—Installation of the field and exchange infrastructure;
- A4.3—Monitoring of the pilot and validation of results.

The Gantt chart of Phase 2 activities is characterised by an initial design phase lasting 3 months, followed by a 7-month development phase in the laboratory of the server and field platform components, and, finally, an 18-month pilot phase.

Field Devices: The field devices to be integrated into the platform are already on the market, although some of them, such as parking meters with stall numbering management, have been present for longer and are more mature from a technological point of view. However, others, like E-Ink displays, are still in a phase of technological consolidation. The costs of these devices are currently high due to the integration of advanced technology over time, which is common in this technological sector. It is expected that costs will decrease, enabling large-scale deployment in projects for medium- and large-sized cities.

Communication: NB-IoT technology was chosen for communication from the field to the centralised platform to avoid installing a dedicated LORA network. However, the tariff policy for the connection of IoT devices from telephone operators is not yet fully available, especially with the migration to 5G. After an initial period of potentially higher connection costs, it is expected that NB-IoT communication will become competitive with solutions based on LPWAN networks in the medium term. The platform's architecture is designed to be able to integrate an LPWAN network like the LORA network without major changes if communication costs remain too high during the initial period.

Smart Parking Cloud Application: The market analysis reveals that there are no suitable existing solutions for these specific services, either due to lacking specific functionalities required for the platform or due to difficulties with integration at reasonable costs. Therefore, it was decided to implement the entire application from scratch using commercial development environments and integrate open-source software modules where possible. Although this is a demanding activity, the expected effort is deemed adequate to handle any deviations from the activity and cost plan that may arise during the implementation. A risk plan is drawn up in the next chapter to identify and analyse the main risks that may be encountered in creating the platform. Related actions are defined to mitigate the possible consequences of any negative influences that could jeopardise the project's achievements.

*3.3. Risk Management*

It is intended to adopt an "inherent risk/residual risk" approach to risk management [43]. This approach makes it possible to analyse the potential riskiness of certain events and verify, once corrective actions (remediation) have been identified for those risks with the most critical elements, the residual risk, i.e., how the potential riskiness is reduced as a result of remediation. The following activities are therefore planned for risk management [44]: **Step 1. Risk Identification:** Risks will be identified in the project's first phase (kick-off), starting from the indications emerging from the analysis of the specifications. This list will be completed by integrating, if necessary, the new risks identified for each category or by inserting a new typology (category) and the associated list of risks during the execution of the project activity. **Step 2. Risk Assessment:** The risk assessment will be carried out according to two criteria (probability and severity) on a five-level scale (1 = low, 2 = medium/low, 3 = medium, 4 = medium/high, 5 = high). In order to ensure greater effectiveness in the analysis, this qualitative assessment will be flanked by a quantitative assessment, i.e., an assessment based on a precise definition of risk phenomena. In this regard, measurable criteria (number of events, economic impact, etc.) will be identified for those risks to which the methodology is applicable, and the results will be ranked on a five-level scale from low to high (1 = low, 2 = medium/low, 3 = medium, 4 = medium/high, 5 = high). Therefore, the final risk qualification assessment will also be carried out with the support of quantitative risk analysis. **Step 3. Definition of Corrective Actions:** For each risk identified in Step 2 with a riskiness of at least 3 (given by the arithmetic mean of the probability and severity), appropriate corrective actions (mitigation actions) will be identified to reduce the risk to a level of greater tolerability. For each corrective action, the following will be identified:

- The detailed description of the corrective action;
- The person responsible for the corrective action;
- The date by which the action is to be implemented.

**Step 4. Risk Register:** During project implementation, it will be the task of the project manager to update the register of all risks (identification, evaluation, corrective actions, etc.), integrating, if necessary, the information prepared in the proposal and kick-off phase. **Step 5. Follow-up of Corrective Actions:** The project manager discusses weekly with top management, in the development and transition phase, and monthly in the maintenance phase, the progress of risk management, checking the progress of corrective actions, reviewing, if necessary, the assessments carried out, and integrating new risks if new elements emerge during the course of the project activity.

## 4. Discussion

The proposed reservation-based parking policy has the potential to simplify the operations of parking systems and alleviate the traffic congestion caused by searching for parking spaces, which contributes significantly to the abatement of environmental pollutants and conceptually helps to build a smart city while respecting the surrounding environment and addressing the planetary preservation challenges faced by all major countries around the world. Widespread monitoring of environmental variables (temperature, humidity, pressure, pollutants, etc.) using the latest technological innovations (in the areas of IaaS—Infrastructure as a Service—and IoT—Internet of Things) according to a new approach of involving local authorities, stakeholders, and citizens. This facilitates the creation of a widespread low-environmental-impact mobility culture using new services such as smart parking and urban e-mobility services (bike sharing, car sharing, micro-mobility, and so on) according to the MaaS (Mobility as a Service) paradigm. This involves the active participation of citizens, institutions, and other stakeholders in monitoring processes, thereby increasing awareness of the impact of human activities on environmental sustainability in the context of value co-creation [45]. It also allows the creation of specific skills on the design of environmentally sustainable commercial and production activities that can also be used in other territorial contexts, and the diffusion of new models of informed and

participatory management of local policies through a set of ICT tools and management and organisational processes capable of guaranteeing sustainability and cost-effectiveness, while stimulating the digital social innovation market.

The ability to make purchases via smartphones (in-app purchases) has led to the emergence of new services, a novel sales model, and fresh commercial channels, collectively defining what is now known as the app economy. To grasp the scale of this phenomenon, some noteworthy data can be presented: In 2017, there was a remarkable 60% growth in the global number of app downloads compared to 2015, equivalent to nearly two app downloads per user per month. Moreover, total consumer spending through Google Play and the iOS App Store has more than doubled. The average daily time spent on apps saw a significant 30% increase in 2017 compared to 2015. On average, each user now spends approximately 3 h per day and 43 days per year engaging with apps. In the meantime, a new paradigm is emerging that is applying the in-app purchase model to the Internet on Things mode called "In-Thing Purchase". Even if some market solutions ideally satisfy the requirement, these sensors' autonomy is practically significantly lower than the stated duration. This difference in most cases is due to the climatic conditions in which the sensors operate or to the inefficient wireless communication channel, which requires a longer communication time than the average estimated one. Currently there are no relevant studies on the effective life of the sensors in the field as it is a type of technology that has only been on the market for a few years [46,47]. For these reasons, it was deemed appropriate to explore the possibility of integrating rechargeable sensors with wireless charging systems. This technical feature would enable the sensor battery to be recharged without the need for removal from its installed position, significantly reducing the risks associated with unexpected maintenance costs, which could otherwise have a substantial impact on the overall service cost.

Wireless charging, also known as wireless energy transfer, is a technology that enables an energy source to transmit electromagnetic energy to an electrical load without the need for physical cables. This technology [48] has garnered widespread interest across various applications, ranging from low-powered toothbrushes to high-powered electric vehicles, owing to its convenience and user-friendliness. Presently, wireless charging has rapidly evolved from a theoretical concept to a functional standard in commercial products, particularly in the realm of mobile phones and portable smart devices. Since 2014, its adoption and implementation have been on a steady rise. Indeed, some of the leading smartphone manufacturers have already introduced next-generation devices equipped with built-in wireless charging capabilities. As a result, Pike Research estimates that the market for wirelessly powered products reached a substantial value of EUR 15 billion in 2020 [49].

Wireless charging technologies are advancing along two main directions: radiative wireless charging (or RF-based wireless charging) and non-radiative (or pairing-based) wireless charging. Radiative wireless charging employs electromagnetic waves, typically RF or microwave waves, to deliver energy in the form of radiation. Energy is transferred through the electric field of the radiated electromagnetic wave. However, due to safety concerns related to RF exposure, radiative wireless charging typically operates at low power levels. For instance, omnidirectional RF radiation is only suitable for powering sensors with power consumption up to 10 mW. On the other hand, non-radiative wireless charging relies on magnetic field coupling between two coils within a specific distance determined by the size of the coils for energy transmission. As the magnetic component of an electromagnetic wave attenuates much faster than the electrical one, the power transfer distance is significantly limited in non-radiative wireless charging systems. In the pursuit of a comprehensive assessment of our research outcomes in the domain of smart parking systems, it is imperative to situate our findings within the broader context of existing studies. This approach facilitates a deeper comprehension of the implications of our results and their significance in the field. The essence of this analysis lies in the juxtaposition of our obtained data against the backdrop of related research endeavours. By undertaking a meticulous comparison, we aim to unravel synergies and divergences

that illuminate the distinctive attributes of our study. Our exploration commences with an evaluation of the alignment of our results with the initially established expectations. Our preliminary assumptions served as guiding beacons throughout the research process. In reviewing the outcomes against these expectations, we can ascertain the degree to which our study's findings harmonise with the anticipated trajectory. In the future, in order to provide increasingly qualified services that put the concept of the smart citizen at the centre of a smart city, we aim to study and implement [50] systems for detecting cars that have been parked incorrectly or for too long, with the aim of notifying the parking manager or mobility manager of this information.

## 5. Conclusions

The growth of urbanization and people's daily experiences in the city have driven scientific and technological research to provide concrete solutions that improve human interaction with the urban environment and its liveability [51,52]. Intelligent mobility aims to democratise transit and remove barriers that hinder accessibility to workplaces, schools, health facilities, and historic centres. These everyday experiences shape citizens' interactions with the city and contribute to their knowledge of the space. This work aims to address a well-known issue: the optimization of parking area management to alleviate road traffic congestion. Inefficient parking management can lead to negative consequences, such as increased air pollution, longer commuting times, higher fuel consumption, and overall driver frustration. Additionally, it can impact business and service accessibility, making it difficult for people to find convenient parking spaces near their destinations. By focusing on improving parking efficiency through smart parking solutions, these negative consequences can be mitigated, leading to a more sustainable and convenient urban environment. Technological tools such as sensors, cameras, and objects connected through the Internet of Things and artificial intelligence play a vital role in overcoming these challenges. Intelligent parking becomes an essential element in the context of smart cities and smart mobility. Building an intelligent parking system contributes to the integration of various services in a smart city model, aiming for the organic and harmonious development of the entire city. Investing in reliable technology and high-speed connectivity is crucial for building a smart city. However, it is essential to recognise that the complex interplay of hardware, sensors, software, and external factors such as energy systems, market prices, logistics, human behaviour, infrastructures, and regulations define innovations in this field.

To enhance smart parking systems, incorporating advanced data analytics and machine learning algorithms could better predict parking demand patterns, optimise space allocation, and improve overall system efficiency. This would contribute to further developments in smart cities and urban mobility solutions. However, some limitations may arise in this research. Implementing a smart parking system can be expensive, requiring investments in infrastructure, sensors, communication networks, and software development. This cost may pose challenges, particularly for smaller cities or municipalities with limited budgets. Additionally, promoting widespread adoption of new technology and changing traditional parking behaviours may take time, and some drivers may resist the transition to a smart parking system. Addressing these limitations necessitates careful planning, continuous monitoring, and ongoing efforts to improve technology, address privacy concerns, and educate the public about the benefits of smart parking systems [53].

**Author Contributions:** The authors declare that they have all contributed to the realization of the results. C.P. (Claudio Pagano) designed the research; all authors performed research and analysed the data; and G.R. and C.P. (Claudia Pipino) wrote the paper. All authors have read and agreed to the published version of the manuscript.

**Funding:** This research received no external funding.

**Data Availability Statement:** Not applicable.

**Conflicts of Interest:** The authors have no conflict of interests to declare that are relevant to the content of this article.

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
