# Peer review of "An Overview of Urban Mobility: Revolutionizing with Innovative Smart Parking Systems"

_sustainability, doi:10.3390/su151713174_

Round 1

Reviewer 1 Report

1.     Explain the meaning of the parentheses in the lines 140, 344

2.     Does the risk assessment phase only contemplate those inherent to the operation of the system? Are financial risks not included?

3.     The technological feasibility of the project seems viable. However, an approximate analysis of its economic feasibility is recommended.

4.     It is also convenient to quantify its effects on the environment such as particle pollution, traffic hours saved, vehicles in waiting lines and others.

5. Finally, it is suggested to the authors to specify if this project is on the table for discussion or if its development and implementation.

A good command of the English language, good writing style

Author Response

We are truly grateful for your valuable advice in order to enhance the quality of our work. It is of great importance to us to address the revisions you have provided, as this has allowed us to produce something that can actively contribute to the scientific community. In particular we can answer point by point to your suggestion: 

i. Error, it has been changed.
         ii. This article does not include an analysis of financial risks. However, this could serve as an idea for a new study, aimed at delving into this aspect for potential future improvements.
         iii. The analysis of technical-economic feasibility was conducted by separately examining the individual blocks that make up the logical architecture of the platform (field devices, communications, and server software modules). 
         iv. added and a new citation has been inserted
         v. The data resulting from this study and the architecture proposed is very close to a real existing architecture. We have just specyfied that is a feasibility study. 

Reviewer 2 Report

Very glad to review this paper (sustainability-2556909). Thanks for your waiting. For the optimization of parking area management to alleviate road traffic congestion, this paper proposed the design of a smart parking architecture, which is able to offer new services by exploiting the latest Internet of Things technologies. In general, this paper is innovative in the design of the smart parking system, but there are many serious problems. First, this paper is more like a project proposal rather than an academic paper. It is more in theory analysis of the feasibility of the design, while the possibility of its implementation is very doubtful. Second, the division of the modules is not reasonable enough, and the division of the paragraphs is very chaotic, resulting in the connection between the context is not tight, and there being no logic. Finally, there are too many detail problems in the text, and the description of the same content appears repeatedly.

 Main problems:

i.              In Section 2, the second half of the fourth paragraph, the fifth paragraph, and the sixth paragraph both explain the flaws of Smart Parking Systems. I do not understand why it should be divided into three paragraphs.

ii.              When introducing different sensors, why did you separate them into two separate paragraphs?

iii.              In Section 3, introductions about ultra-low power information displays and wireless charging systems are missing. You seem to put some missing pieces in the last two paragraphs of Section 4.

iv.              An explanation of Figure 1 is missing.

v.              The interpretation of Figure 1 and Figure 2 is not clearly distinguished.

vi.              Section 4 begins with “ The results of... ”, but I can't see anything about the experiment in this paper.

Minor problems:

vii.              There are two periods at the end of line 30, one of which should be deleted.

viii.              There are empty parentheses on line 122 and line 144. What is the reference?

ix.              The front part of Section 2.1 is repeated above and is not recommended to be elaborated again.

x.              The first letter of the first word of the sentence that begins in the second half of line 239 is not capitalized.

xi.              " ]. " in " [24,25].]. " of line 250 is superfluous.

xii.              What is the content of Table 3? Why is there only "Table 3" in this paper?

xiii.              In all tables, abbreviations such as "Pros" and "Cons" are not recommended for the header line.

xiv.              Following the format above, line 274 and line 384 should not be indented.

xv.              In line 299, there is a sentence "such as the optical sensor, which detects the change in light." In line 230, the working principle of the optical sensor has been explained in detail, so the phrase "which detects the change in light" can be deleted.

xvi.              Why is the introduction to On-Line Booking Systems beginning from line 320 not in a separate paragraph, which has nothing to do with the introduction of sensors above?

xvii.              What does the " [??] " in line 344 mean?

xviii.              Why the line 335 not indented?

xix.              The sharpness of the two pictures is not high enough, it is blurred after magnification, and the width of the two pictures is set slightly larger.

xx.              In Section 3.4, are steps 2 and 4 both risk assessments? There's ambiguity here.

The language level is clear but still need the authors to check more. 

Author Response

We are truly grateful for your valuable advice in order to enhance the quality of our work. It is of great importance to us to address the revisions you have provided, as this has allowed us to produce something that can actively contribute to the scientific community. As for the major revisions, certain sections of the manuscript have been restructured, aiming to closely to the suggestions provided. Particularly, responses to each point are outlined below:
    MAJOR
        i. error with latex paragraph. It has been fixed in one paragraph. 
        ii. error with latex paragraph. Fixed now 
        iii. the item list at the begin of the paragraph in sec 3 has been adeguated to the explaination. No missing pieces now.
        iv. fixed
        v. fixed
        vi. fixed and it has been adeguated. 
    MINOR:
        vii. fixed, deleted one period
        viii. two error, there aren't citation in those lines. Fixed.
        ix. done by creating one section in order to be not confused
        x. fixed.
        xi. fixed
        xii. latex writing error. Fixed
        xiii. fixed. 
        xiv. fixed.
        xv. done. 
        xvi. latex error in generating paragraph. Fixed.
        xvii. fixed.
        xviii. this should not be indented in list item. It has been created a new paragraph
        xix. the dimensions have been reduced to evict the blurring.
        xx. fixed, one has been deleted. 

Reviewer 3 Report

The study proposes a design of an intelligent parking architecture. This is an interesting and novel topic. Thus, it presents important practical applications, especially in the transportation and mobility sector. In the following, I will make some recommendations that I consider can improve the quality of the manuscript.

The introduction and literature review are quite complete. The existing literature has been adequately synthesized, contextualizing the reader on the problems addressed in the study.

The methodology and results are clearly and adequately presented.

However, I consider that the discussion should be further developed. This section should explain and contrast the data obtained with other research in this field of study. Thus, the authors should answer questions such as: were the results in accordance with expectations, are the results congruent with other similar research? And, if not, what elements explain the discrepancies that have occurred?

I also recommend including a specific section with the limitations of the study and future lines of research.

Author Response

We are truly grateful for your valuable advice in order to enhance the quality of our work. It is of great importance to us to address the revisions you have provided, as this has allowed us to produce something that can actively contribute to the scientific community. The results and discussion section has been expanded as suggested, aiming to address the suggested improvements to the best of our ability. In particular, a new segment has been added that places greater emphasis on comparing the data generated during the literature review phase with the data resulting from the analysis of the proposed architecture. We sincerely appreciate the valuable advice provided and the insightful review that has contributed to the enhancement of our work.

Round 2

Reviewer 2 Report

Thanks for authors' revision. I can see the all concerns are well addressed and the paper is improved much. Thus, allow me to give a positive recommendation. 

The language level of the revised paper is acceptable.